# Crop Type Identification Using High-Resolution Remote Sensing Images Based on an Improved DeepLabV3+ Network

Zhu Chang [1], Hu Li [1,*,†], Donghua Chen [1,2,†], Yufeng Liu [2], Chen Zou [1], Jian Chen [1], Weijie Han [3], Saisai Liu [2] and Naiming Zhang [2]

[1] School of Geography and Tourism, Anhui Normal University, Wuhu 241003, China; 2021011468@ahnu.edu.cn (Z.C.); chendonghua@chzu.edu.cn (D.C.); zouchen192@ahnu.edu.cn (C.Z.); 2021011465@ahnu.edu.cn (J.C.)
[2] School of Computer and Information Engineering, Chuzhou University, Chuzhou 239000, China; liuyufeng@chzu.edu.cn (Y.L.); lss@chzu.edu.cn (S.L.); zhangnaiming@chzu.edu.cn (N.Z.)
[3] Anhui National Defense Science and Technology Information Institute, Hefei 230041, China; 2021011470@ahnu.edu.cn
\* Correspondence: lihu2881@ahnu.edu.cn; Tel.: +86-183-2537-6353
† These authors contributed equally to this work.

**Abstract:** Remote sensing technology has become a popular tool for crop classification, but it faces challenges in accurately identifying crops in areas with fragmented land plots and complex planting structures. To address this issue, we propose an improved method for crop identification in high-resolution remote sensing images, achieved by modifying the DeepLab V3+ semantic segmentation network. In this paper, the typical crop area in the Jianghuai watershed is taken as the experimental area, and Gaofen-2 satellite images with high spatial resolutions are used as the data source. Based on the original DeepLab V3+ model, CI and OSAVI vegetation indices are added to the input layers, and MobileNet V2 is used as the backbone network. Meanwhile, the upper sampling layer of the network is added, and the attention mechanism is added to the ASPP and the upper sampling layers. The accuracy verification of the identification results shows that the MIoU and PA of this model in the test set reach 85.63% and 95.30%, the IoU and F1_Score of wheat are 93.76% and 96.78%, and the IoU and F1_Score of rape are 74.24% and 85.51%, respectively. The identification accuracy of this model is significantly better than that of the original DeepLab V3+ model and other related models. The proposed method in this paper can accurately extract the distribution information of wheat and rape from high-resolution remote sensing images. This provides a new technical approach for the application of high-resolution remote sensing images in identifying wheat and rape.

**Keywords:** crop identification; improved DeepLab V3+ network; GaoFen-2; Jianghuai watershed

## 1. Introduction

As social and economic development, rapid population growth, and the impact of global climate change and urban expansion continue, sustainable agricultural development faces significant challenges [1,2]. Therefore, obtaining timely, efficient, and accurate information on the planting area and spatial distribution of crops is crucial for government management in formulating agricultural food policies, optimizing land resources, adjusting agricultural planting structures, and ensuring national food security [3,4].

Satellite remote sensing technology, with its advantages of wide coverage and high timeliness, has become one of the primary approaches for identifying crop types [5–8]. In recent years, experts and scholars have conducted extensive research on the extraction of wheat and rape planting areas based on satellite remote sensing data. Most of these studies have focused on medium- to low-resolution satellite remote sensing data, such as SPOT/HRV [9], MODIS [10,11], Landsat [12,13], and Sentinel [14–18]. Medium- to low-resolution remote sensing data provide rich spectral information and high temporal

resolution. The rich spectral information enhances the separability between different land covers, while the high temporal resolution enables accurate mapping of crop distribution by capturing the spectral variations during different phenological stages of crops. However, wheat and rape are often planted in scattered patterns, and the field areas of rape are typically small. The limitations of spatial resolution in medium- to low-resolution remote sensing data restrict the accuracy of identifying wheat and rape [19–21]. The use of high-resolution satellite imagery significantly enhances the spatial resolution of crop features in images, reducing the occurrence of mixed pixels. This better meets the requirements of precision agriculture management [22,23]. However, high-resolution satellite imagery has weaker spectral information and lower temporal resolution. Weak spectral information, especially in areas with fragmented plots and complex planting structures, makes it difficult to solve the problem of "homologous substances having disparate spectra, and heterologous substances having the same spectrum" [24,25]. In order to obtain more precise information about the cultivation areas of crops from high-resolution satellite imagery, scholars have delved into the research of remote sensing identification of crops using deep learning methods.

Current research indicates that for remote sensing data sources with weak spectral information and high spatial resolution, deep learning methods can capture deeper semantic features in images. This effectively improves the problem of low accuracy in land cover classification caused by the phenomenon of "homologous substances having disparate spectra, and heterologous substances having the same spectrum" [26–30]. This can mitigate the issue of "homologous substances having disparate spectra, and heterologous substances having the same spectrum". Among them, Atrous Spatial Pyramid Pooling (ASPP) proposed by the DeepLab semantic segmentation model series can enlarge the receptive field without changing the resolution and fuse features at different levels. It achieves a good performance in object boundary segmentation. DeepLab V3+ combines the advantages of the Encoder-Decoder (ED) structure and the ASPP module, making it currently one of the most outstanding semantic segmentation algorithms in terms of comprehensive performance [31–35]. However, due to its multiple down-sampling operations, the DeepLab V3+ network is prone to losing boundary information of land covers, which reduces the recognition accuracy. Therefore, it is of great significance to explore how to better protect the edge information of wheat and rape fields and improve the recognition accuracy while using the DeepLab V3+ network in order to promote the application of deep learning models in high-resolution remote sensing for wheat and rape identification.

In summary, to enhance the accuracy of crop identification from high-resolution remote sensing images, this paper uses Gaofen-2 (GF-2) satellite imagery as the data source and proposes an improved method for crop identification using the DeepLab V3+ network. The proposed method modifies the input layer of the traditional DeepLab V3+ network to better distinguish between wheat, rape, and other land covers; adds upper sampling layers to alleviate the problem of boundary information loss; and increases an attention mechanism to highlight the information of crops in the network layers, thereby improving the recognition accuracy of wheat and rape. Our study extends the application of high-resolution satellite remote sensing data in precision agriculture management, which could provide technical support for the agricultural sector in fine management.

## 2. Materials and Methods

### 2.1. Study Area

The study area is located in Dingyuan County, Chuzhou City, Anhui Province (117°11′~117°31′, 32°26′~32°42′), which belongs to the North subtropical humid monsoon climate zone with an annual average temperature of 15 °C (Figure 1). Dingyuan County is located in the Jianghuai Watershed region, which marks the boundary between northern and southern China. It is a major agricultural planting county in Anhui Province, where wheat and rape are the main crops in spring, while rice and soybeans are the main

crops in autumn. The study area is characterized by three types of landforms: hills, undulating plains, and plains. The agricultural planting structure is complex, with both collective planting in plain areas and scattered distribution in mountainous areas. Due to the influence of terrain and climate, the remote sensing imaging of crops can easily form the phenomenon of "homologous substances having disparate spectra".

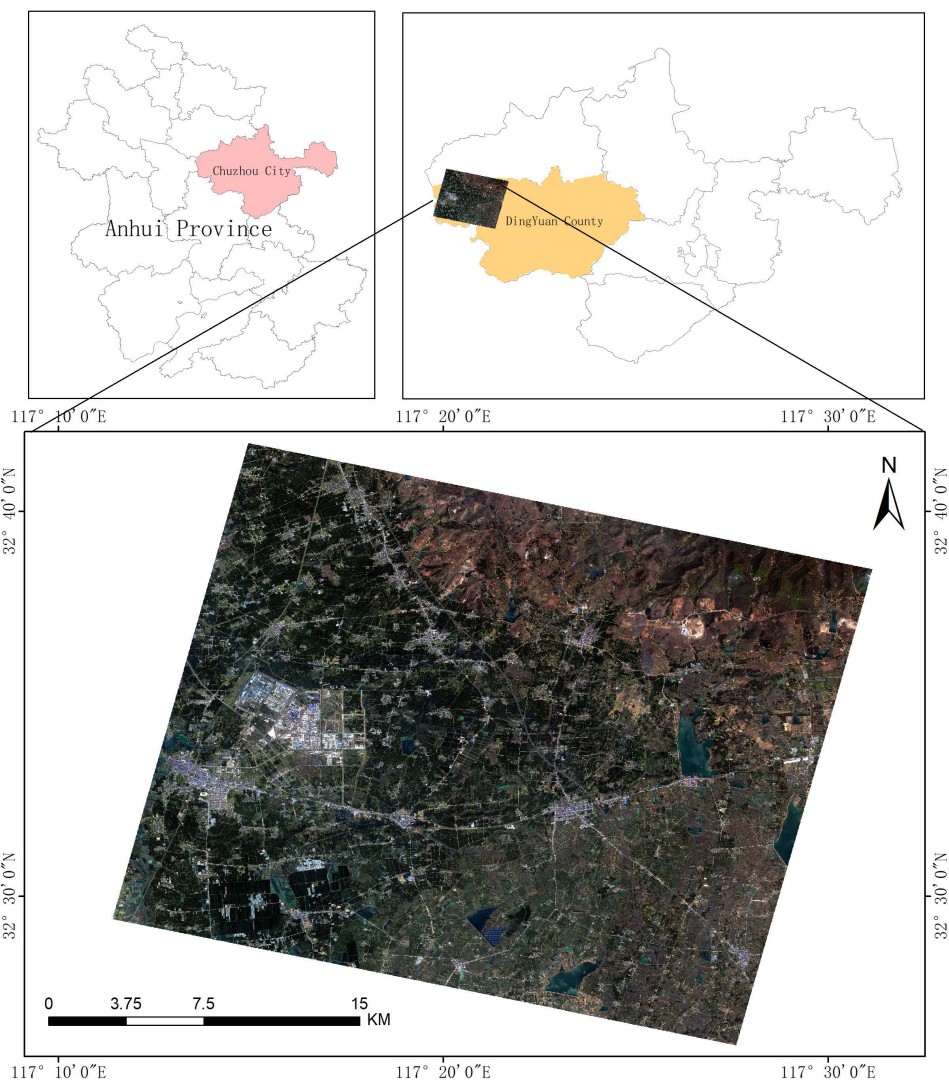

**Figure 1.** Schematic Diagram of Study Area.

## 2.2. Data Source and Preprocessing

### 2.2.1. Source and Preprocessing of Remote Sensing Data

In this study, we selected GF-2 PMS2 data acquired on 3 April 2022, which corresponded to the flowering stage of rape and the jointing stage of wheat. The original GF-2 PMS2 image consisted of a multi-spectral image with a spatial resolution of 4 m and a panchromatic image with a spatial resolution of 1 m. To eliminate radiation errors caused by atmospheric scattering, we conducted radiation calibration and atmospheric correction on the multi-spectral image. Orthographic correction was also employed to correct the geometric deformation of the multi-spectral images. On the other hand, radiometric calibration and orthographic correction were applied to correct the errors caused by transducers and geometrical deformation of the panchromatic image, respectively. Finally, the ortho-corrected multi-spectral image and panchromatic image were fused to produce a multispectral image with a spatial resolution of 1 m.

### 2.2.2. Construction of Data Set

The quality and quantity of the sample set are crucial factors that affect the accuracy of deep learning. In this study, we obtained the sample set through manual pixel-by-pixel annotation by combining field investigation data, Sentinel-2 time-series data and GF-2 imagery were acquired during the flowering stage of rape. This approach allowed us to create a robust and diverse sample set that was suitable for effectively training the deep learning model.

The GF-2 imagery clearly illustrates the spectral and textural discrepancies between rape and wheat during the rape flowering period. By combining this information with field sampling data, most of the planting areas can be identified manually. However, certain rape and wheat areas may exhibit characteristics in the imagery that deviate from those typical of the normal rape flowering period due to factors such as climate, topography, and planting time. To address this issue, this study employs time-series Sentinel-2 data to identify unknown areas using the spectral changes in rape and wheat during various phenological stages.

In this study, Level-2A Sentinel-2 data are obtained from the Google Earth Engine (GEE) platform. Figure 2 illustrates the images of wheat and rape acquired on 25 February, 7 March, 6 April, 11 April, 21 April, 6 May, and 15 June, respectively. The figure clearly shows that the spectral features of the wheat and rape planting areas vary over time. Thus, incorporating time-series Sentinel-2 data as ancillary information can aid in accurately identifying the correct planting range of wheat and rape in GF-2 imagery.

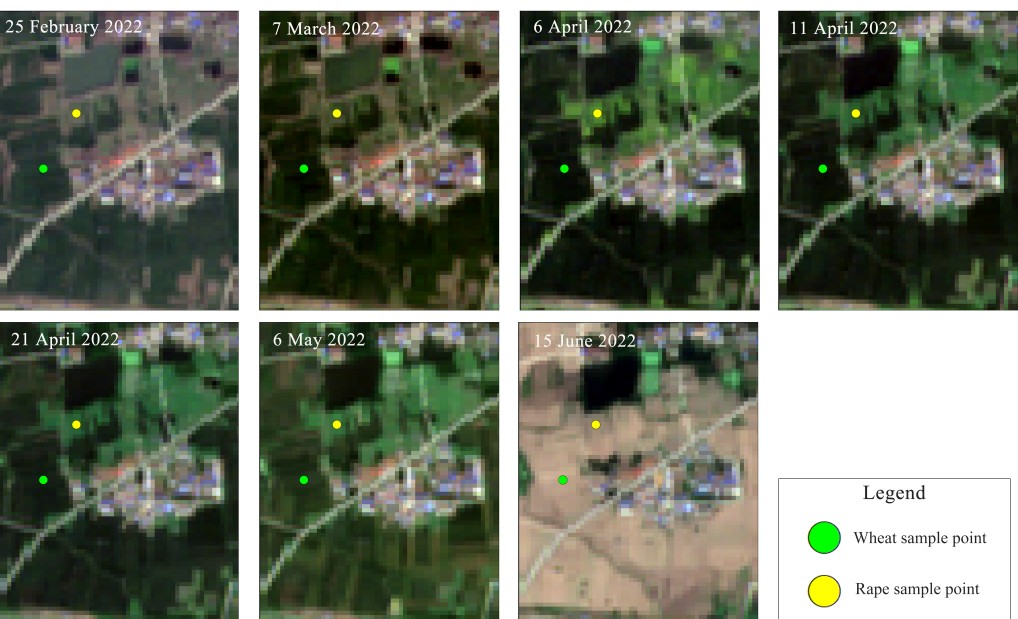

**Figure 2.** Sentinel-2 image of wheat and rape in different periods.

This study selected 10 areas within the study area based on field survey data, as well as the temporal Sentinel-2 data and GF-2 imagery during the flowering period of rape, to create the sample set. Using GF-2 images as the base map, we manually annotated each pixel of the image for wheat, rape, and other land cover types using ArcGIS software, with all other land cover types uniformly classified as background. After the manual annotation process, the vector data were converted to raster data with the same resolution as the GF-2 imagery, and wheat was assigned a pixel value of 3, rape was assigned a pixel value of 2, and other land cover types were assigned a pixel value of 1. Out of the ten selected areas, seven were used to create the training and validation sets while the remaining three were used for the testing set. A Python program was developed to crop the images and corresponding labels to a size of 128 × 128 pixels and augment the resulting 4242 image-

label pairs used for training and validation by flipping and rotating them 90 degrees. This process ultimately generated a dataset of 12,726 image-label pairs, which were divided into training and validation sets at a 4:1 ratio. The test images and corresponding labeled samples used in this study are shown in Figure 3.

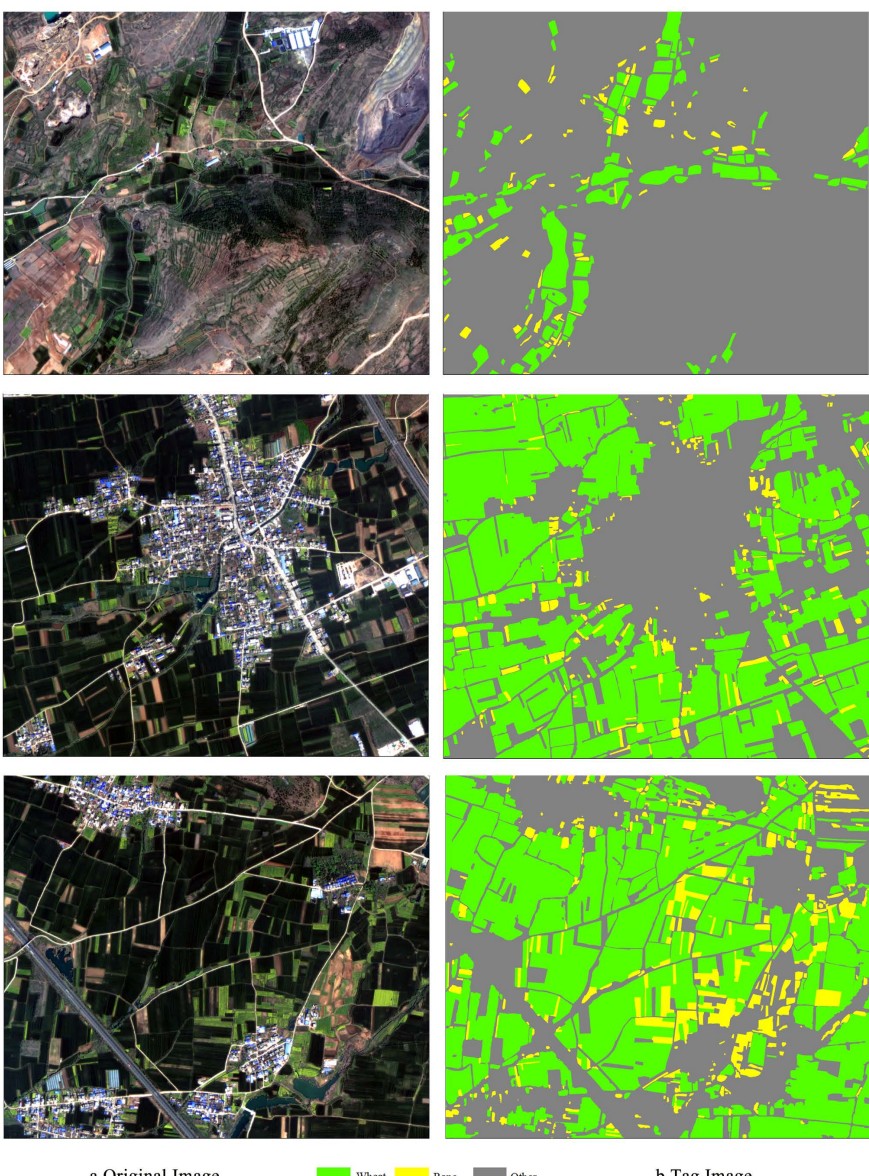

a.Original Image     <span style="color:green">■</span> Wheat   <span style="color:yellow">■</span> Rape   <span style="color:gray">■</span> Other     b.Tag Image

**Figure 3.** Corresponding images and actual annotated samples in the test set. (**a**) Original image corresponding to the three regions of the test set; (**b**)Tag image corresponding to the three regions of the test set.

### 2.3. Vegetation Feature

The prior studies have demonstrated that incorporating vegetation features into the extraction of crop distribution information from remote sensing imagery can result in improved discrimination between vegetation and non-vegetation, as well as different types of vegetation, compared to relying solely on the original four spectral bands [36–38]. This study employed prior knowledge to extract four additional vegetation indices from the original four spectral bands, namely the Normalized Difference Vegetation Index (NDVI), Green Normalized Difference Vegetation Index (GNDVI), Optimal Soil Adjusted Vegetation Index (OSAVI), and Canola Index (CI) [17,39]. The calculation formulas of the four vegetation indices are shown in Table 1.

**Table 1.** Formula for calculating vegetation index.

| Vegetation Index | Calculation Formula | |
|---|---|---|
| NDVI | $NDVI = \dfrac{NIR - R}{R + NIR}$ | (1) |
| GNDVI | $GNDVI = \dfrac{NIR - G}{G + NIR}$ | (2) |
| OSAVI | $OSAVI = 1.16 * \dfrac{NIR - R}{1.16 + R + NIR}$ | (3) |
| CI | $CI = NIR * (G + R)$ | (4) |

To evaluate the performance of the four indices in distinguishing different vegetation types in the study area, this study extracted the values of wheat, rape, and forest corresponding to the sample points in the four index feature maps and compared the differences in these values. As shown in Figure 4, the frequency distributions of extracted values from the four vegetation indices for wheat, rape, and forest sample points are presented. The peaks of wheat, rape, and forest in the NDVI are approximately 0.73, 0.55, and 0.52, respectively; the peaks in the GNDVI are approximately 0.68, 0.52, and 0.47, respectively; the peaks in the OSAVI are approximately 0.84, 0.63, and 0.59, respectively; the peaks in the CI are approximately $0.60 \times 10^7$, $1.20 \times 10^7$, and $0.45 \times 10^7$, respectively. Based on the overlap of the curves and the distribution of peaks, it can be concluded that NDVI, GNDVI, and OSAVI can well distinguish between wheat and rape, wheat and forest; CI index can well distinguish between rape and wheat, rape and forest.

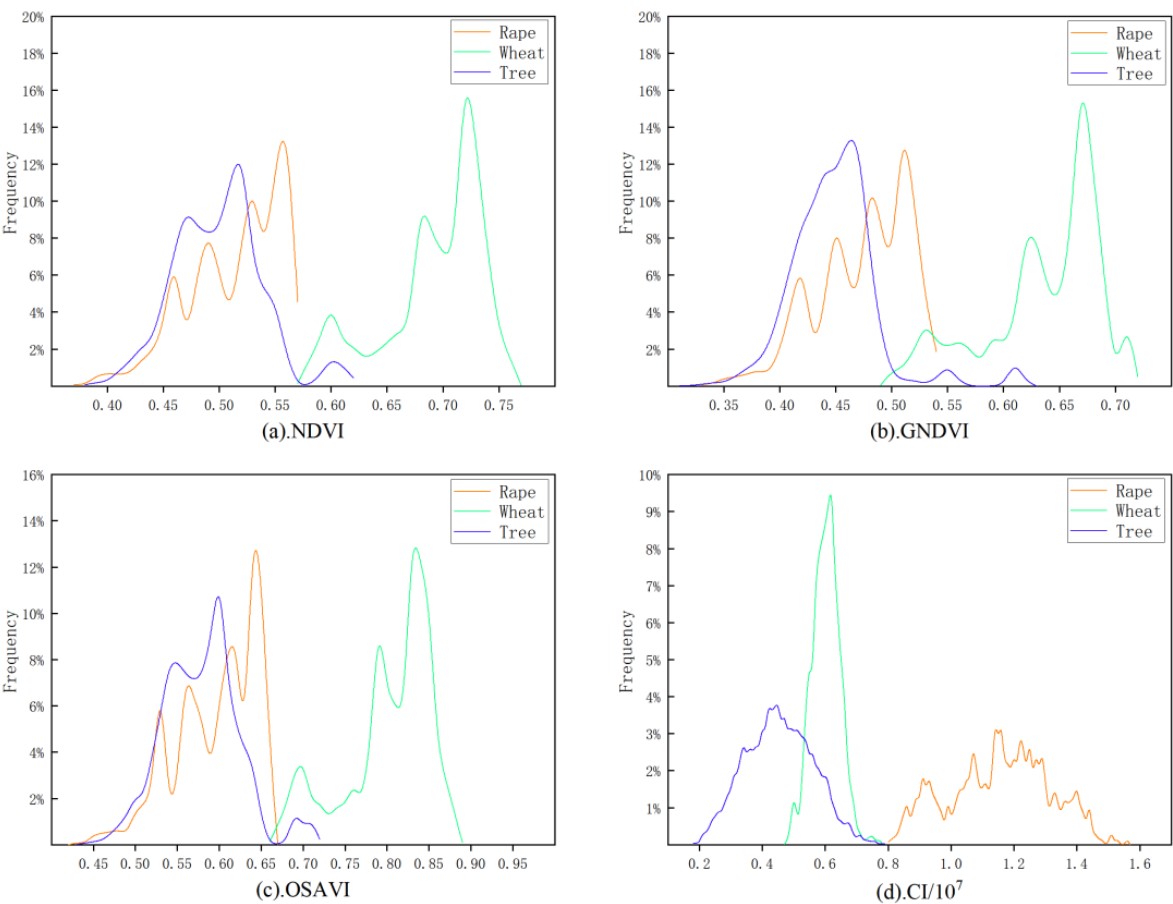

**Figure 4.** Frequency distribution of different vegetation in different indices. (**a**) Frequency distribution of three plantings in NDVI; (**b**) frequency distribution of three plantings in GNDVI; (**c**) frequency distribution of three plantings in OSAVI; (**d**) frequency distribution of three plantings in CI.

### 2.4. Improved DeepLab V3+ Classification Method

DeepLab V3+ is a typical semantic segmentation model which has achieved excellent results in the remote sensing field. Although the DeepLab V3+ model encodes rich semantic information, it also has some shortcomings when it is directly applied to wheat and rape recognition in high spatial resolution remote sensing images [33,40]. In order to make the model more suitable for wheat and rape identification by remote sensing technology, this paper makes the following improvements on the basis of DeepLab V3+. Firstly, vegetation indices were added to the input layer to enhance the differentiation among ground objects. To reduce computation and memory usage and improve computation speed, the Xception network in DeepLab V3+ was replaced with the lightweight MobileNet V2 network. Secondly, the down-sampling multiple was set to 8 and an additional upper sampling layer was added to mitigate the loss of edge information of wheat and rape due to multiple down-sampling. To increase sensitivity to the wheat and rape regions, a Convolutional Block Attention Module (CBAM) was added to the ASPP module and the upper sampling layer. Finally, the weighted cross-entropy loss function was introduced to address the issue of sample imbalance among wheat, rape, and other ground objects. The network architecture is shown in Figure 5.

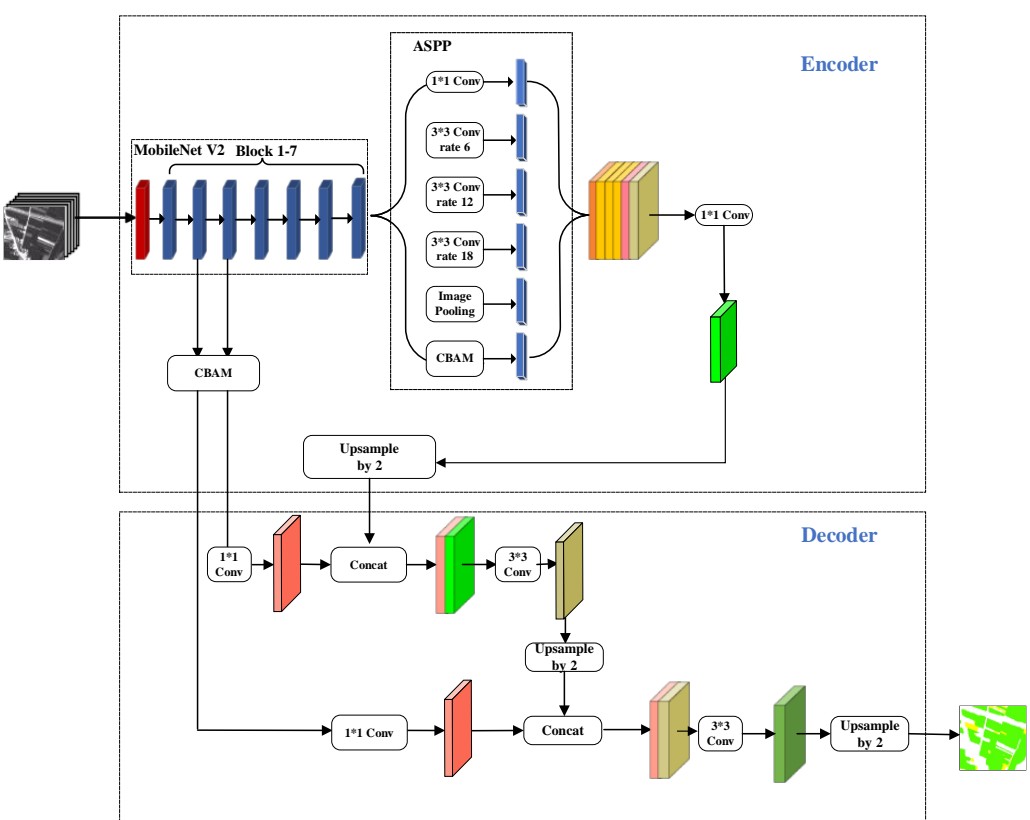

**Figure 5.** Improved DeepLab V3+ network structure diagram.

### 2.5. CBAM Module

Attention mechanisms have been shown to be crucial in human perception as they allow us to focus on the most salient parts of a scene and capture its visual structure more effectively. In recent years, researchers have added attention mechanisms to network models to improve their recognition performance [41–48]. One such mechanism is the CBAM, which combines both channel and spatial attention modules. By incorporating the CBAM module into a convolutional layer, important features in both the channel and spatial domains can be highlighted, leading to more accurate recognition results. The CBAM structure diagram is shown in Figure 6.

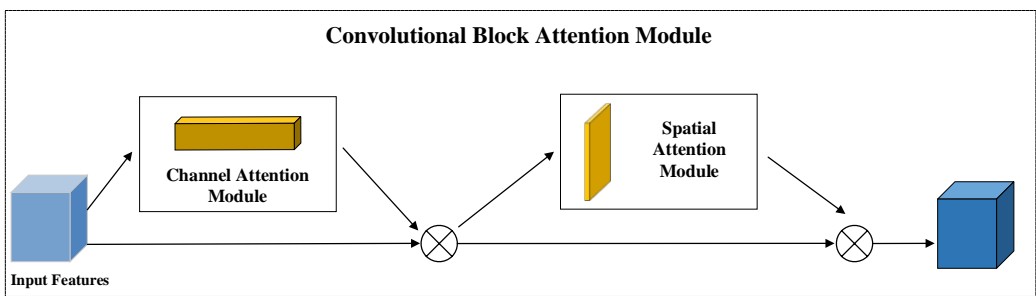

**Figure 6.** Schematic diagram of CBAM structure.

The purpose of the spatial attention module is to highlight the spatial location distribution that is more important for recognition in the convolutional layer. The specific operation is as follows: Two pooling methods, global average pooling and global maximum pooling, were used to compress the channel of input feature graph F to obtain $F_{avg}^s$ and $F_{max}^s$, respectively. The two feature graphs were combined and reduced to one channel by a $7*7$ convolution operation. Then, the weight graph of spatial attention was obtained by Sigmoid function. The spatial attention weighted graph $F_{Sout}$ is obtained by multiplying the obtained weight graph with the original feature graph F. Figure 7 shows the structure of the spatial attention mechanism, and its formula is as follows:

$$F_{Sout} = \sigma\left(f^{7*7}\left(F_{max}^s \oplus F_{avg}^s\right)\right) \otimes F \tag{5}$$

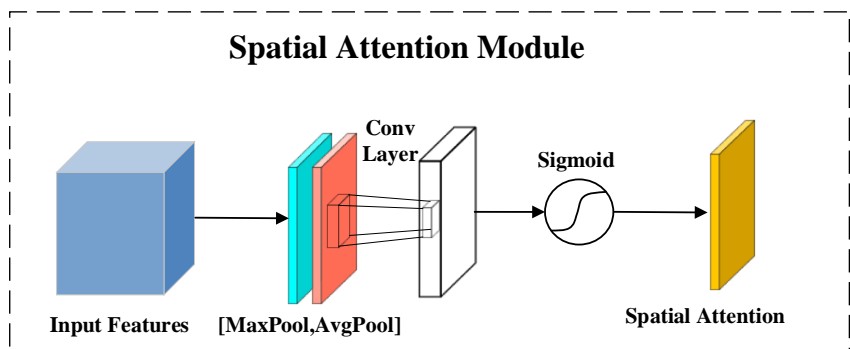

**Figure 7.** Schematic diagram of spatial attention mechanism structure.

In the formula, $\sigma$ represents the Sigmoid function, $f^{7*7}$ represents the $7*7$ convolution operation, $\oplus$ represents the channel to merge, $\otimes$ represents elements multiplication.

The purpose of the channel attention module is to highlight the channels of great value for recognition in the convolutional layer. The specific operations are as follows: global average pooling and global maximum pooling are used to compress the spatial dimension of the feature map F to obtain $F_{max}^c$ and $F_{avg}^c$. Then, input $F_{max}^c$ and $F_{avg}^c$ into the multi-layer perceptron MLP, respectively, and add the obtained results. Input the added results into the Sigmoid function to generate the weight of channel attention. Multiply the generated weight with the original feature graph F to obtain the channel attention weighted graph $F_{Cout}$. Figure 8 shows the mechanism structure of channel attention, and its formula is as follows:

$$F_{Cout} = \sigma\left(MLP(F_{max}^c) + MLP(F_{avg}^c)\right) \otimes F \tag{6}$$

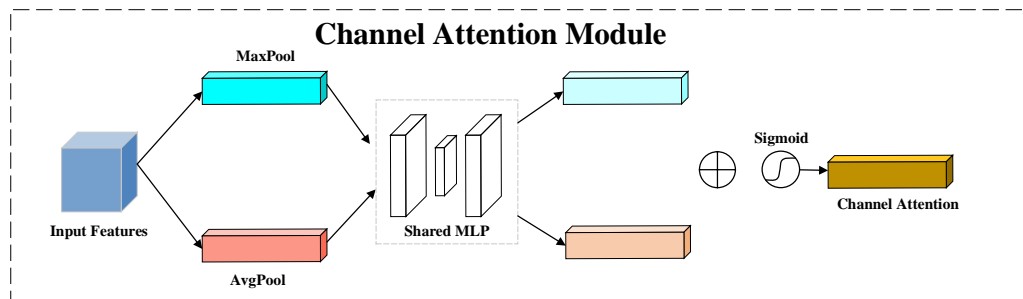

**Figure 8.** Schematic diagram of channel attention mechanism structure.

### 2.6. Weighted Cross-Entropy Loss Function

The cross-entropy loss function is used as the semantic segmentation loss, which measures the similarity between the ground truth and predicted values. The traditional cross-entropy function assigns the same weight to each class of recognition objects, which may result in low training efficiency when one class of target samples has a significantly larger quantity than other classes [40,49]. Due to the small number of rape samples and the large number of background samples in the dataset used in this study, the network tends to learn the features of the background. To improve the problem of low segmentation accuracy caused by imbalanced sample quantities in the dataset, a weighted cross-entropy loss function is introduced. Based on the original cross-entropy loss function, different weights are assigned to wheat, rape, and background classes. The specific formula is shown as follows:

$$Loss = -\frac{1}{N} \sum_{n=0}^{N-1} \sum_{c=0}^{C-1} w_c y_{n,c} \log(p_{n,c}) \tag{7}$$

In the formula, N is the total number of pixels, C is the total number of categories, n represents the nTH training pixel, c represents the category of training pixel, $y_{n,c}$ is the real value of the nTH pixel, $p_{n,c}$ is the predicted value of the nTH pixel, $w_c = \frac{N-N_c}{N}$ represents the weight of category c, and $N_c$ represents the total number of pixels of category c.

### 2.7. Model Training

This study utilized the NVIDIA GeForce RTX2060 SUPER graphics card to accelerate the experiments, and the PyTorch framework was employed to construct the model. During the training process, the Adam optimizer function was selected as the parameter optimizer, with a batch size of 32 and a total of 200 epochs. To enhance the training efficiency, the learning rate with dynamic attenuation was adopted, and the initial value was set to 0.01.

To assess the effectiveness of semantic segmentation, researchers typically rely on commonly used evaluation metrics. These include Pixel Accuracy (PA), Mean Intersection over Union (MIoU), Precision, Recall, F1_Score, Intersection over Union (IoU), among others. In this study, PA and MIoU were utilized to evaluate the overall classification accuracy of the model. Meanwhile, F1_Score and IoU were chosen to evaluate the recognition accuracy of wheat and rape.

PA is used to assess the prediction accuracy of each pixel in image classification. The formula for calculation is as follows:

$$PA = \frac{\sum_{i=0}^{k} p_{ii}}{\sum_{i=0}^{k} \sum_{j=0}^{k} p_{ji}} \tag{8}$$

IoU is used to assess the overlap between each predicted class and the corresponding ground truth in an image after semantic segmentation. The formula for calculation is as follows:

$$IoU = \frac{p_{ii}}{\sum_{j=0}^{k} p_{ij} + \sum_{j=0}^{k} p_{ji} - p_{ii}} \tag{9}$$

MIoU represents the mean IoU of all classes, and the calculation formula is as follows:

$$\text{MIoU} = \frac{1}{k+1} \sum_{i=0}^{k} \frac{p_{ii}}{\sum_{j=0}^{k} p_{ij} + \sum_{j=0}^{k} p_{ji} - p_{ii}} \tag{10}$$

F1 Score is a comprehensive metric for evaluating the precision and recall of each class in classification results. Precision assesses the accuracy of each class in the classification results, while recall evaluates the completeness of each class in the classification results. The formulas for precision, recall, and F1 Score are as follows:

$$\text{Precision} = \frac{p_{ii}}{\sum_{j=0}^{k} p_{ji}} \tag{11}$$

$$\text{Recall} = \frac{p_{ii}}{\sum_{j=0}^{k} p_{ij}} \tag{12}$$

$$F1_{Score} = \frac{2 * Recall * Precision}{Recall + Precision} \tag{13}$$

In the formulas for the above evaluation metrics, $k$ represents the number of classes, $p_{ii}$ represents the number of pixels correctly identified as class $i$, $p_{ji}$ represents the predicted value for class $i$, $p_{ij}$ represents the actual value for class $i$.

## 3. Results and Analysis

### 3.1. Effects of Different Vegetation Indices on Identification Results

To assess the efficacy of various vegetation indices in extracting wheat and rape from GF-2 images captured during the flowering stage, this study employed an improved DeepLab V3+ network as the classification model. The four vegetation indices were added to the input layer of the model separately (the initial input layer only included the original four bands). The resulting precision performance of the model on the test set after the addition of each vegetation index is presented in Table 2.

**Table 2.** Add the identification accuracy of different vegetation features.

| | MIoU | PA | IoU | | F1_Score | |
|---|---|---|---|---|---|---|
| | | | **Wheat** | **Rape** | **Wheat** | **Rape** |
| Origin | 82.14% | 93.69% | 91.97% | 68.47% | 95.82% | 81.37% |
| +NDVI | 84.68% | 94.67% | 92.78% | 73.41% | 96.28% | 84.78% |
| +GNDVI | 84.33% | 94.56% | 92.71% | 72.10% | 96.22% | 83.74% |
| +OSAVI | 84.95% | 94.65% | 92.71% | 73.75% | 96.22% | 84.94% |
| +CI | 84.96% | 94.64% | 92.71% | 74.17% | 96.25% | 84.99% |

Based on the experimental results, it is evident that adding various vegetation indices based on the initial four bands enhances both the overall accuracy and the identification accuracy of wheat and rape. Notably, the classification accuracy of the model is notably influenced by different vegetation indices. The MIoU and PA values obtained using CI and OSAVI are significantly higher than those obtained using NDVI and GNDVI, indicating that the addition of CI and OSAVI improves the overall classification accuracy of the model better than NDVI and GNDVI. Furthermore, CI leads to the best classification accuracy of rape, followed by OSAVI, both of which surpass NDVI and GNDVI. On the other hand, the effect of vegetation index on the classification accuracy of wheat is negligible. In conclusion, the experiment establishes that adding CI and OSAVI significantly improves the recognition accuracy of the model, followed by NDVI, while GNDVI has the smallest impact on the recognition accuracy of the model.

To enhance the recognition accuracy of wheat and rape and further improve the performance of the model, this study added CI, OSAVI, NDVI, and GNDVI to the input layer in a sequential manner based on their impact on the recognition results, following the previous experiments. Table 3 presents the recognition accuracy of the model when two, three, and four vegetation indices were sequentially added to the input layer.

**Table 3.** The identification accuracy of vegetation index with different quantity is added.

|  | **MIoU** | **PA** | **IoU** | | **F1_Score** | |
|---|---|---|---|---|---|---|
|  |  |  | **Wheat** | **Rape** | **Wheat** | **Rape** |
| +CI | 84.96% | 94.64% | 92.71% | 74.17% | 96.25% | 84.99% |
| +CI+OSAVI | 85.63% | 95.30% | 93.76% | 74.24% | 96.78% | 85.51% |
| +CI+OSAVI+NDVI | 84.27% | 94.67% | 92.78% | 73.41% | 96.28% | 84.78% |
| +CI+OSAVI+NDVI+GNDVI | 84.33% | 94.41% | 92.45% | 72.67% | 96.07% | 84.31% |

Based on the experimental results, it can be concluded that adding only CI and OSAVI to the input layer results in the best overall recognition accuracy of the model, as well as the recognition accuracy of wheat and rape. Specifically, the MIoU and PA reached 85.63% and 95.30%, respectively, while the IoU and F1_Score for wheat and rape were 93.76% and 96.78%, and 74.24% and 85.51%, respectively. However, adding three or four vegetation indices did not lead to a further improvement in the recognition accuracy of the model, and it may increase the computer memory consumption. Therefore, in this study, CI and OSAVI were selected to be added to the original four bands as the feature set for the experiments in this paper and were used as the input layer for the model. In the following experiments, the input layer was always the feature set including CI and OSAVI.

In order to further evaluate the impact of the feature set constructed in this paper on the model performance, we recorded the changes in Loss and MIoU on the validation set during each iteration of the model in the training process. As depicted in Figure 9, we compared the Loss and MIoU curves in the validation set during model training using two different input layers. The Origin is only using the original four bands as the input layer, the Origin+VI is adding CI and OSAVI on the basis of the original four bands, with a total of six bands as the input layer.

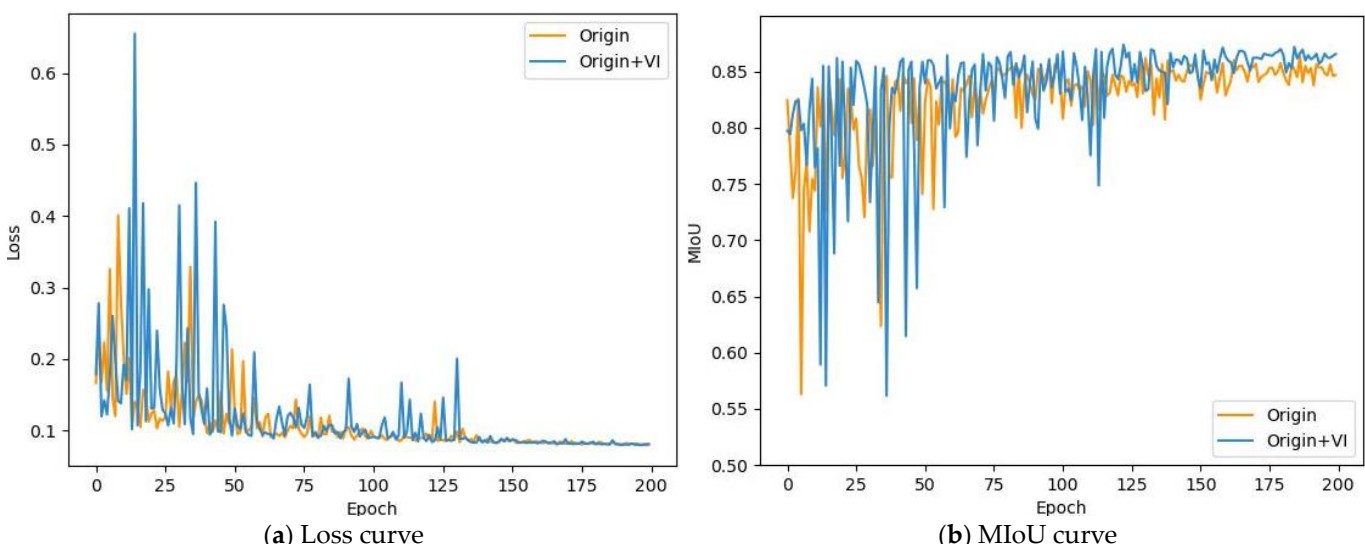

(**a**) Loss curve                    (**b**) MIoU curve

**Figure 9.** Loss curve and MIoU curve correspond to different input layers. (**a**) Loss curve corresponding to different input layers; (**b**) MIoU curve corresponding to different input layers. Origin represents using only the original four bands as the input layer, while Origin+VI represents adding two vegetation indices, OSAVI and CI, to the original four bands as the input layer.

The analysis of the loss and MIoU curves during the model training process indicates that the convergence rate of the model slows down when the vegetation index features are added, but convergence can still be reached before 200 iterations, and the loss value after convergence has only a minor difference between the two input layers. On the other hand, the MIoU curve clearly shows that adding the vegetation index features leads to a significant improvement in the model's accuracy on the verification set. Thus, it can be concluded that the feature set proposed in this paper has a positive impact on the performance of the model.

### 3.2. Ablation Experiment

To further verify the effectiveness of the improved DeepLab V3+ model in this study, ablation experiments were conducted based on the original DeepLab V3+ model. Following the improvement method proposed in this paper, five schemes were developed, which are outlined below:

Scheme 1: DeepLab V3+ model based on Xception as backbone network;

Scheme 2: DeepLab V3+ model based on Mobilenet V2 as backbone network;

Scheme 3: CBAM is added to the upper sampling layers and ASPP layer based on DeepLab V3+ structure of Scheme 2;

Scheme 4: Add the upper sampling layer based on DeepLab V3+ structure of Scheme 2;

Scheme 5: Based on DeepLab V3+ structure of Scheme 2, add CBAM to the upper sampling layers and ASPP layer, and add the upper sampling layer (MyDeepLab V3+).

This paper compared the effects of different backbone networks on model performance in Scheme 1 and Scheme 2. The results, as shown in Table 4, indicate that when the backbone network was replaced from Xception to lightweight MobileNet V2, the number of model parameters was reduced from 208.7 MB to 27.91 MB, which is about 7.5 times smaller, greatly reducing the consumption of computer memory. Training time was also reduced, with the duration of an epoch of the model decreasing from 265 ms to 100 ms. Comparing the forecast time of a single image, MobileNet V2 reduced the forecast time of a single image from 10 ms to 4 ms. Furthermore, the model's overall accuracy MIoU was improved by 4.6% by modifying the backbone network. Therefore, MobileNet V2 was selected as the backbone network in this paper as it not only reduced memory consumption and improved training speed, but also improved the recognition speed and accuracy of the model.

**Table 4.** Comparison of model parameters and identification accuracy of different backbone networks.

| Backbone Network | MIoU | PA | Parameter Size/MB | Training Times/Epoch | Single Graph Prediction Time/ms |
|---|---|---|---|---|---|
| Xception | 84.16% | 94.51% | 208.7 | 265 | 10 |
| Mobilenet V2 | 84.62% | 94.71% | 27.91 | 100 | 4 |

In order to compare the effect of adding attention mechanisms and increasing upper sampling on model accuracy, this paper designed four schemes, namely Scheme 2, Scheme 3, Scheme 4, and Scheme 5, to conduct comparative tests. Figure 10 shows the convergence of the MIoU and Loss curves for these schemes on the verification set within 200 epochs. The proposed method (Scheme 5) achieved the highest average test accuracy and the highest accuracy under the convergence state. The accuracy of the models with only the CBAM added (Scheme 3) and only the upper sampling layer added (Scheme 4) were lower than that of the proposed method but slightly higher than that of the original model (Scheme 2). The experimental results demonstrate that adding the CBAM module or the upper sampling layer to the original DeepLab V3+ model can effectively improve the identification accuracy of the model. Furthermore, adding both the CBAM and the upper sampling layer can significantly improve the performance of the model for crop type recognition.

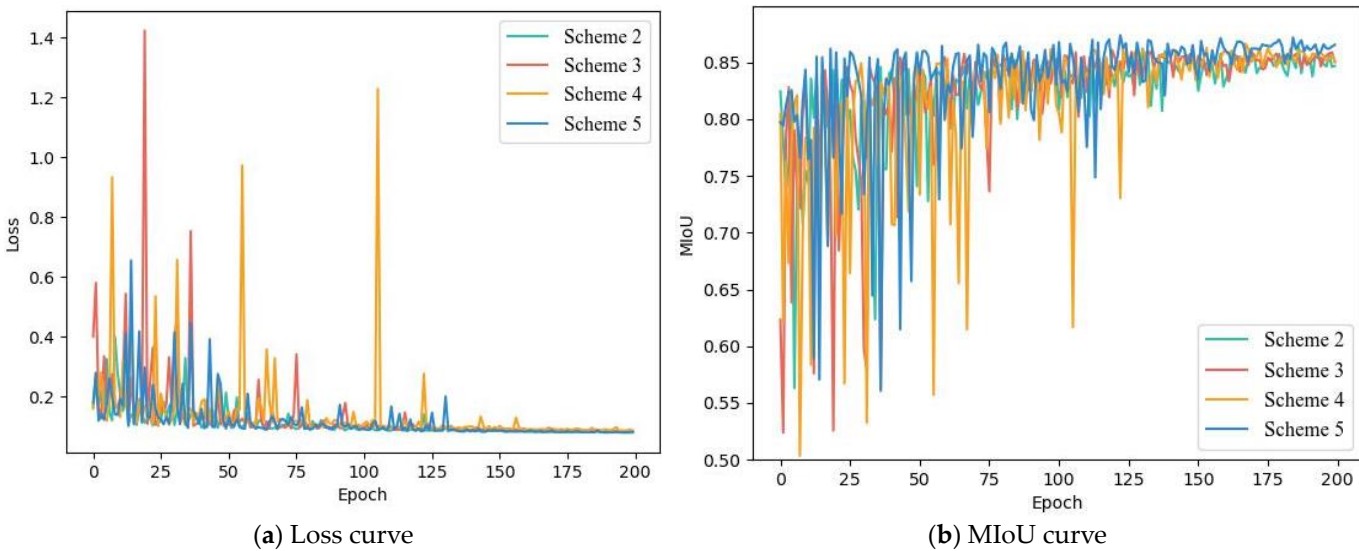

(**a**) Loss curve          (**b**) MIoU curve

**Figure 10.** Loss curve and MIoU curve corresponding to different comparative experiments. (**a**) Loss curve corresponding to different comparative experiments; (**b**) MIoU curve corresponding to different comparative experiments. Scheme 2: DeepLab V3+ model based on Mobilenet V2 as backbone network; Scheme 3: CBAM is added to the upper sampling layers and ASPP layer based on DeepLab V3+ structure of Scheme 2; Scheme 4: Add the upper sampling layer based on DeepLab V3+ structure of Scheme 2; Scheme 5: Based on DeepLab V3+ structure of Scheme 2, add CBAM to the upper sampling layers and ASPP layer, and add the upper sampling layer (MyDeepLab V3+).

### 3.3. The Influence of Different Semantic Segmentation Models on Recognition Results

To demonstrate that the improved DeepLab V3+ model in this study has certain advantages in using remote sensing technology to identify wheat and rape, its crop identification performance is compared with that of widely used models in crop identification using remote sensing technology, including SegNet, U-Net, and original DeepLab V3+. The performance of different models was evaluated in two aspects: (1) analyzing the accuracy and convergence of the loss value of each model on the validation set; (2) comparing the mean intersection over union (mIoU), pixel accuracy, wheat and rapeseed intersection over union (IoU), and F1 score of each model on the test set.

Figure 11 shows the Loss curve and MIoU curve on the verification set in the training process of SegNet, U-Net, DeepaLab V3+, and MyDeepLab V3+. It can be observed that the Loss and MIoU curves of the four models exhibit diverse trends. All four models' Loss curves show a generally declining pattern and reach convergence within 200 epochs, indicating satisfactory convergence. Notably, MyDeepLab V3+ shows the fastest convergence rate, suggesting superior training effectiveness. On the other hand, the MIoU curves of the four models showed different trends. The variation trend of the MyDeepLab V3+ model is more stable than that of the SegNet and U-Net models, and the average test accuracy and convergence accuracy of the MyDeepLab V3+ model are the highest among the four models. Therefore, through the Loss and MIoU analysis of the validation sets during the training of the four models, it can be concluded that MyDeepLab V3+ is the best model to identify wheat and rape among the four models.

To evaluate the generalization ability of four models, the recognition accuracy of wheat and rape obtained by the four models in the test set was compared, and the MIoU, PA, IoU, and F1_Score of wheat and rape obtained by the four models on the test set were also calculated. As shown in Table 5, the overall accuracy (MIoU, PA) and the accuracy of wheat and rape recognition (IoU, F1_Score) of MyDeepLab V3+ proposed in this paper were the highest among SegNet, U-Net, DeepLab V3+, and MyDeepLab V3+. Specifically, compared with SegNet, U-Net, and DeepLab V3+, the proposed method improved the MIoU by 3.36%, 1.65%, and 1.01%, respectively. In terms of wheat recognition accuracy, the

proposed method had an IoU improvement of 1%~2% compared to the other three models, and the F1_Score improvement was less than 1%. In terms of rape recognition accuracy, the proposed method had an IoU improvement of 7.03%, 4.03%, and 2.10% compared to SegNet, U-Net, and DeepLab V3+, respectively, and an F1_Score improvement of 5.05%, 3.03%, and 1.64%. Through the comparison of these indicators, it can be concluded that the MyDeepLab V3+ model proposed in this paper has a higher model generalization performance and more accurate recognition of the planting range of wheat and rape.

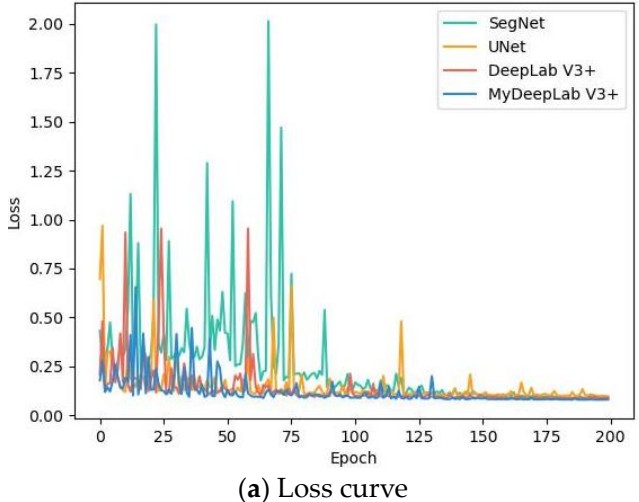 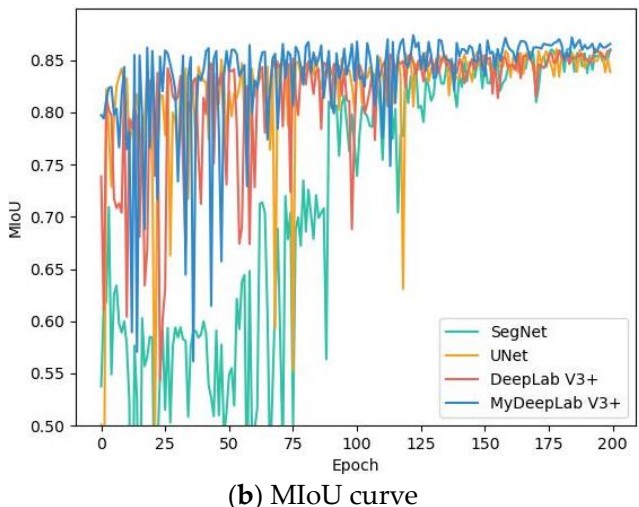

(**a**) Loss curve          (**b**) MIoU curve

**Figure 11.** Loss curve and MIoU curve correspond to different semantic segmentation models. (**a**) Loss curve corresponding to different semantic segmentation models; (**b**) MIoU curve corresponding to different semantic segmentation models.

**Table 5.** Comparison of recognition accuracy of different semantic segmentation models.

| | MIoU | PA | IoU | | F1_Score | |
|---|---|---|---|---|---|---|
| | | | Wheat | Rape | Wheat | Rape |
| SegNet | 82.27% | 94.12% | 92.04% | 67.21% | 95.85% | 80.46% |
| UNet | 83.98% | 94.69% | 92.92% | 70.21% | 96.33% | 82.48% |
| DeepLab V3+ | 84.62% | 94.71% | 92.81% | 72.14% | 96.27% | 83.87% |
| MyDeepLab V3+ | 85.63% | 95.30% | 93.76% | 74.24% | 96.78% | 85.51% |

*3.4. Result Analysis*

According to the previous description, the MyDeepLab V3+ model proposed in this paper achieves high recognition accuracy and extraction performance in identifying the planting areas of wheat and rape using remote sensing technology. In the three validation areas, including mountainous regions, planting areas around urban areas, and areas with a relatively high concentration of wheat and rape planting, the model can accurately identify the planting areas of wheat and rape, demonstrating good generalization ability (Figure 12). For the recognition results of wheat, the model is basically consistent with the actual planting areas, but there is still room for improvement in the detailed recognition of some small ridges. For the recognition results of rape, the model can accurately identify the planting areas without excessive errors or omissions, but there is some discrepancy in the preservation of the boundary information of rape plots compared to actual planting. Overall, the model shows good recognition performance in practical applications and can provide important support for agricultural production.

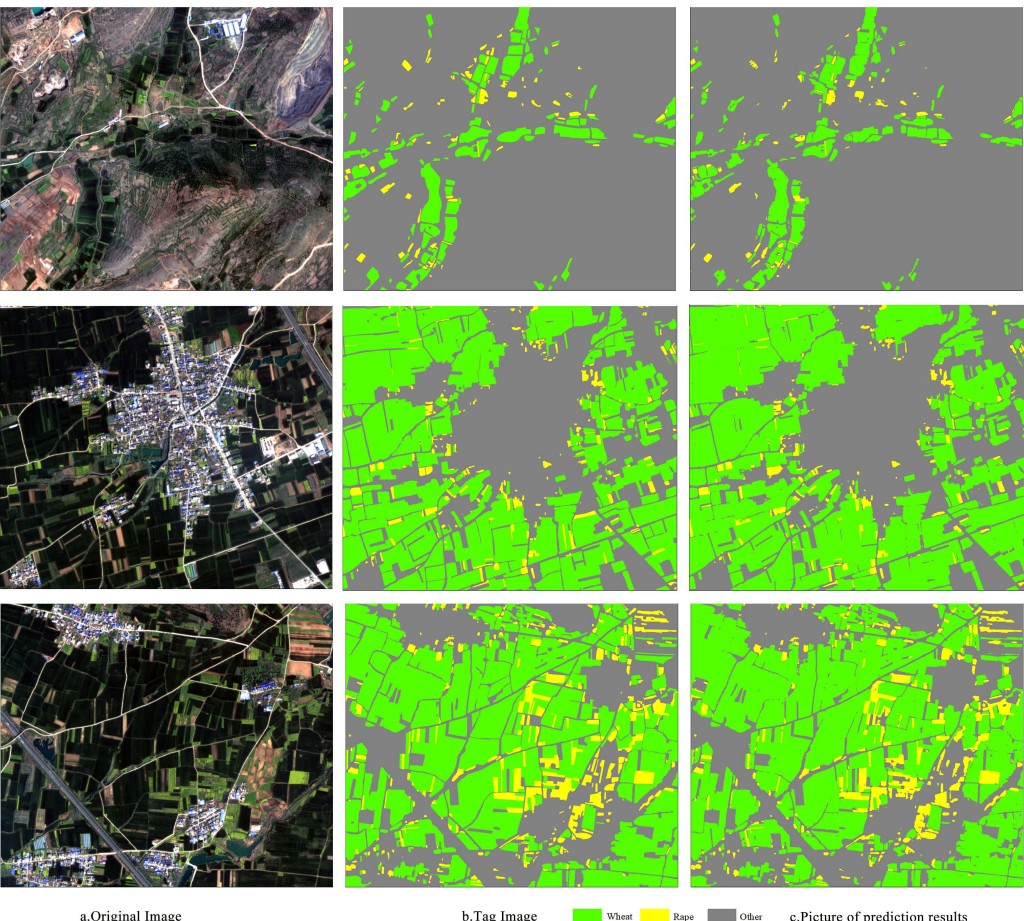

a.Original Image          b.Tag Image    ■ Wheat   ■ Rape   ■ Other      c.Picture of prediction results

**Figure 12.** Comparison chart of identification results. (**a**) Original images of the test set; (**b**) real labels for the test set; (**c**) the results of model recognition in this article. The three areas with concentrated testing are mountainous regions, planting areas around urban areas, and areas with a relatively high concentration of wheat and rape planting, respectively.

## 4. Discussion

### 4.1. Advantages of the Algorithm in This Paper

This study proposes an improved method for crop identification using high-resolution satellite imagery from the GF-2 satellite. In contrast to the traditional DeepLab V3+ model, modifications are made to both the input layer and the network structure. Experimental results validate the effectiveness of these enhancements, demonstrating a significant improvement in crop recognition accuracy in the modified model.

Tian [17] and Ashourloo [39] introduced the temporal CI for rape identification and conducted comparative experiments to validate its superior performance over conventional vegetation indices such as NDVI. To assess the impact of using only the rape flowering period CI index on crop identification accuracy when employing high-resolution remote sensing imagery, a series of comparative experiments were conducted. As shown in Figure 13, the IoU for rape was obtained when different indices were incorporated. Experimental results indicate that integrating CI into the input layer of deep learning effectively improves the accuracy of rape identification. The recognition performance surpasses that achieved by incorporating only NDVI, GNDVI, and OSAVI.

With the continuous development of deep learning, introducing attention mechanisms into models has proven effective in improving recognition accuracy. Wang [43] successfully enhanced the discriminative capability of the ecological environmental elements in the Yangtze River source region by incorporating CBAM into the ASPP layer. Cai [50] achieved increased precision in the segmentation of camphor leaf spots by adding CBAM to the upsampling

layer. To investigate the impact of CBAM modules in the field of crop recognition, we introduced CBAM separately into the upsampling layer and ASPP layer. Through experimental comparisons, we found that adding CBAM also improves the accuracy of crop recognition.

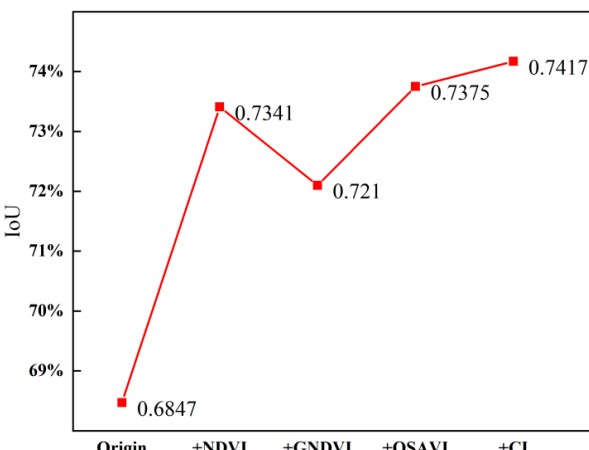

**Figure 13.** The IoU of rape under different vegetation index was added in the input layer. Origin represents the model that only uses the original four bands as input layers. +NDVI represents the model that adds NDVI as an input layer on top of the original four bands. +GNDVI represents the model that adds GNDVI as an input layer on top of the original four bands. +OSAVI represents the model that adds OSAVI as an input layer on top of the original four bands. +CI represents the model that adds CI as an input layer on top of the original four bands.

The traditional DeepLab V3+ model, characterized by multiple downsampling steps, is susceptible to blurring the boundaries of fine-grained features. Xu [49] addressed this limitation by introducing an upsampling layer to the DeepLab V3+ model, effectively boosting the classification accuracy of ground objects in high-resolution remote sensing images. To assess the impact of increasing upsampling layers on crop recognition, we augmented the original model with an additional upsampling layer. Experimental results demonstrate that the inclusion of an upsampling layer significantly improves the accuracy of crop recognition. Furthermore, we validated that simultaneously introducing CBAM while increasing the upsampling layer more efficiently enhances the accuracy of crop recognition. Figure 14 illustrates the recognition accuracy of models improved through different methodologies on the test set.

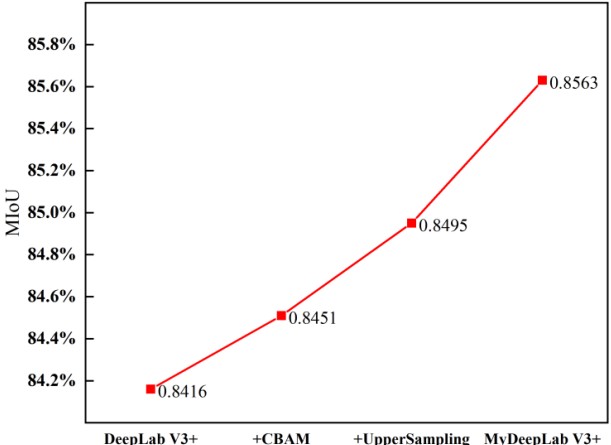

**Figure 14.** The MIoU obtained on the test set by models with different improvement methods. DeepLab V3+ is the original model; +CBAM means adding CBAM to the original model; +UpperSampling means adding an upsampling layer to the original model; MyDeeplab V3+ means adding both CBAM and an upsampling layer to the original model.

### *4.2. Deficiency and Prospect*

The high spatial resolution of the GF-2 satellite renders it well-suited for crop identification, thereby catering to the requirements of precision agriculture. Nonetheless, the constraints of its swath width and revisit cycle led us to employ only one scene image as the experimental area and prevented us from extending the model to crop identification at a county or city level. As a future direction, we plan to explore the potential of GF-1 and GF-6 satellite images, which offer a wider swath width, shorter revisit cycle, and 2 m spatial resolution, as data sources to apply the model to crop identification at a county or city level.

This study focused exclusively on constructing and applying models during the specific phenological period of rape flowering. In future work, we consider integrating the constructed model with the phenological periods of both wheat and rape. This approach aims to enhance the model's applicability in practical agricultural production, considering the challenges associated with acquiring high-quality satellite imagery during the rape flowering period due to weather and imaging constraints.

## 5. Conclusions

In order to enhance the precision of crop identification in regions characterized by fragmented land plots and intricate cropping structures, this study presents an advanced high-resolution remote sensing crop recognition approach based on the DeepLab V3+ semantic segmentation network, utilizing GF-2 satellite imagery as the primary data source. To amplify spectral differences between diverse land features, a feature set is constructed by incorporating the original four bands alongside two vegetation indices, CI and OSAVI. The original Xception backbone network is replaced with the more lightweight MobileNet V2 to reduce network parameters and streamline training time. Additional upsampling layers are integrated into the network, and CBAM modules are introduced in the ASPP module and upsampling layers. These adjustments enhance the model's sensitivity to wheat and rape regions, mitigate the blurring of crop edges due to multiple downsampling, and ultimately elevate crop recognition accuracy. Experimental findings affirm the efficacy of the proposed method, achieving MIoU and PA metrics of 85.63% and 95.30%, respectively, on the test set. Specifically, for wheat, the IoU and F1_Score are 93.76% and 96.78%, while for rape, the IoU and F1_Score are 74.24% and 85.51%, respectively.

**Author Contributions:** Z.C. and H.L. designed the study and wrote the first manuscript draft. Z.C., C.Z., J.C., W.H., S.L. and N.Z. collected data and conducted the statistical analyses. H.L., D.C. and Y.L. revised them critically. All authors have read and agreed to the published version of the manuscript.

**Funding:** We acknowledge the support of Major science and technology projects of high resolution earth observation system (76-Y50G14-0038-22/23,30-Y60B01-9003-22/23); Anhui Province Science and Technology Major Special Project (202003a06020002); Anhui Province Key Research and Development Program Project (2021003, 2022107020028); Anhui Province Special Support Program (2019); Anhui Provincial Outstanding Young Researcher Program for Higher Education Institutions (2022AH020069); Anhui University Collaborative Innovation Project (GXXT-2021-048); Chuzhou City Science and Technology Pro-gram Project (2021ZD013); Natural Science Foundation of Anhui Province (2208085QD107); Natural Science Research Project of Universities in Anhui Province (KJ2021A1063).

**Data Availability Statement:** All data, models, or code generated or used during the study are available from the author at request (2021011468@ahnu.edu.cn).

**Acknowledgments:** We thank the data support of Anhui High Resolution Earth Observation System Data Product and Application Software Research and Development Center.

**Conflicts of Interest:** The authors declare no conflict of interest.

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
