# Peer review of "Crop Type Identification Using High-Resolution Remote Sensing Images Based on an Improved DeepLabV3+ Network"

_remotesensing, doi:10.3390/rs15215088_

Round 1

Reviewer 1 Report

The authors of this manuscript did reasonable literature review but could have more. The algorithm was well introduced. The experiences were designed scientifically and results were well presented.  The discussion and interpretation of results are good.

Overall I accept the manuscript to be published in present form.  

Author Response

     Thank you very much for taking the time out of your busy schedule to revise my paper, it's an honor for me.

     In response to the issue of "adding more literature reviews," I have rephrased the introduction in the first section and increased the number of references to 50. Thank you very much for your valuable suggestion.

     Thank you once again for your careful revisions to the paper. I wish you a happy life and smooth sailing.

Reviewer 2 Report

The terms in the IoU and PA are not clear in the abstract. These must be defined. 

The term sub-meter resolution is not a scientific term to present the resolution. 

Para-1 in Introduction is more generic about the importance of crop type mapping. it can be sequized to maximum 2 sentences by presenting the rationale of the research. 

para-2 in introduction the terms domestic researcher is not a suitable term, regional or local researcher is abetter term. 

Better is to avoid the term etc in scientific writing. 

the authors lacks in the consistancy of the story writing about their objectives. e.g. line 54 and 55. the benefit of high resolution are also presented.

why the last para of the introduction is the complete summary of the paper. it must contains the focused objective/aims of the research.

Research area and data must be placed in material and method section, is it not a part of methodology?

Extended methodology of the paper is presented. 

Discussion is more generic and repetition of the  words like this paper present. it shows the use of an AI tool for writing the paper segment that why the technical story of the paper is missed in this paper. 

the extended conclusion are like the summary of the research work. it must the scientific conclusion drawn from the research work conducted by the authors. avoid unnecessary writing for a technical paper. 

Reviewer 3 Report

Please see the attached minor comments

Satisfactory

Round 2

Reviewer 2 Report

The suggested changes are not found in the response file. only generic terms are provided that the repose is addressed. how it is addressed is not clear to me in the response file. e.g. "In response to the issue that "Para-1 in Introduction is more generic about the importance of crop type mapping. it can be sequized to maximum 2 sentences by presenting the rationale of the research.", I have revised Para-1 in Introduction accordingly. Thank you very much for your valuable suggestion."

Round 3

Reviewer 2 Report

The reviewer sufficient address the suggestions.